# Exclusive breastfeeding among beneficiaries of a nutrition enhancement programme and its associated factors in Ghana

**Martin Nyaaba Adokiya**[1]*, **Mohammed Bukari**[2], **Joyce A. Ndago**[3], **Robert B. Kuganab-Lem**[4], **Humphrey Garti**[2], **Matthew Y. Konlan**[5], **Amata Atinlie Amoasah**[6], **Zakari Ali**[7]

**1** Department of Epidemiology, Biostatistics and Disease Control, School of Public Health, University for Development Studies, Tamale, Ghana, **2** Department of Nutritional Sciences, School of Allied Health Sciences, University for Development Studies, Tamale, Ghana, **3** Department of Social and Behavioral Change, School of Public Health, University for Development Studies, Tamale, Ghana, **4** Institute of Health, Population and Environment Research, Tamale Technical University, Tamale, Ghana, **5** Department of Public Health, Northern Regional Health Directorate, Ghana Health Service, Accra, Ghana, **6** World Food Programme, Tamale, Ghana, **7** Nutrition and Planetary Health Theme, MRC Unit The Gambia at the London School of Hygiene and Tropical Medicine, Serrekunda, The Gambia

* mnyaaba11@uds.edu.gh

**Data Availability Statement:** All relevant data are within the manuscript and its supporting information flies.

## Abstract

### Introduction

Despite the gains on exclusive breastfeeding (EBF), recent nationwide surveys have consistently revealed a decline in EBF rates in Ghana. The World Food Programme implemented an intervention for Enhanced Nutrition and Value Chain (ENVAC) which was based on three pillars including pregnant women, lactating women, adolescent and children under two years old being beneficiaries of the third pillar since the first 1000 days are critical for averting malnutrition. The social behavior change communication (SBCC) interventions implemented as part of this project have a potential to increase EBF among beneficiaries but this has not been measured. Therefore, this study assessed the prevalence of EBF practice among mothers with children under two years old who were beneficiaries of the ENVAC project and its associated factors in northern Ghana.

### Methods

This was a cross-sectional study involving 339 mother-child pairs in two districts of the northern region of Ghana. Participants were mother-child pairs who benefitted from the ENVAC project, which used SBCC strategies to promote good feeding and care practices as well as address other causes of malnutrition during antenatal care and child welfare clinic services among pregnant women, lactating mothers, and children under two years. We used WHO standard questionnaire to assess breastfeeding practices. Factors associated with EBF were modelled using multivariable logistic regression.

### Results

Exclusive breastfeeding was 74.6% (95%CI = 69.5% -79.2%) in the ENVAC project areas, a 31.7% points higher than recent national levels. Adjusted analyses showed that EBF

**Funding:** The authors received no specific funding for this work.

**Competing interests:** The authors have declared that no competing interests exist.

**Abbreviations:** aOR, Adjusted Odds Ratio; BFHI, Baby Friendly Hospital Initiative; CBT, Cash Based Transfers; CHPS, Community-based Health Planning and Services; CI, Confidence Interval; CNF, Complementary Nutritious Foods; CWC, Child Welfare Clinics; EBF, Exclusive Breastfeeding; ENVAC, Enhanced Nutrition and Value Chain; GDHS, Ghana Demographic and Health Survey; GHS, Ghana Health Service; IEC, Information, Education and Communication; IYCF, Infant and Young Child Feeding practices; JHS, Junior High School; MICS, Multiple Indicator Cluster Survey; SBCC, Social Behaviour Change Communication; SHS, Senior High School; WFP, World Food Program; WHO, World Health Organization.

practice was associated with increasing maternal education: moderately educated women [aOR = 4.1 (95% CI = 2.17–7.66), P<0.001], and high [aOR = 9.15, (95% CI = 3.3–25.36), P<0.001], and access to pipe-borne water in households [aOR = 2.87, (95% CI = 1.11–7.43), P = 0.029].

## Conclusion

A social behaviour change communication strategy implemented by ENVAC to lactating mothers likely improved exclusive breastfeeding practice in two districts of northern Ghana. EBF practices were higher among beneficiaries with high education and households with access to pipe-borne water. A combination of SBCC strategies and maternal and household factors are likely the best way to increase EBF rates in impoverished communities and warrants further investigation through future research.

## Introduction

According to the World Health Organization (WHO), breastmilk is the best source of nutrition for infants [1]. WHO recommends that infants should be exclusively breastfed for the first six months of life [2]. Exclusive breastfeeding (EBF) is defined as an infant receiving only breastmilk and no other liquids or solids except for drops or syrups consisting of vitamins, minerals, or medicines [3, 4]. For the first six months of life, breastmilk alone is the ideal nourishment, providing all nutrients including vitamins and minerals an infant need. This implies no other liquid or food is needed for the infant [5]. Breastmilk has been suggested to enhance cognitive development [6], physical, neurological, and as well as protect children against allergies and infectious diseases [7]. However, the common practice is early cessation of breastfeeding in favour of commercial breastmilk substitutes, introduction of liquids such as water and juices.

Globally, 43.5% of infants less than 6 months old were exclusively breastfed in 2020 [8]. In high income countries (e.g. Australia), early initiation of EBF was about 96.0%, this rate declined in the first few weeks of postpartum to only 15.0% and 9.0% of infants being exclusively breastfed at 5 months and 6 months respectively [9]. In Africa, 37% of infants under 6 months of age were exclusively breastfed in 2017 [10]. However, national statistics available in Ghana shows that EBF rates range between 43% and 52% [11, 12]. Early introduction of water and porridges is common practice across Africa. This inhibits EBF practice and exposes infants to disease and nutritional risks [13]. Women in the Sahel have pointed to the high heat index as the reason for giving their infants water [14]. In Ghana, a short duration of maternity leave (3 months) has been cited as a key contributor to early cessation of EBF among working mothers [15]. As women participation in formal employment has continued to grow in recent years in Ghana, for instance the proportion of women in service has grown from 33.5 in 1993 to 54.8 in 2019 [16], there has been a concurrent declines in the number of women who practice EBF for the 6 months recommended period.

Consequently, a number of interventions have been implemented aimed at improving EBF rates in Ghana [17]. These interventions, among other goals, address institutional barriers to EBF and promote safe breastfeeding at health facilities, work places (both public and private). The interventions include the adoption of the 1991 Baby-Friendly Hospital Initiative (BFHI) and the formation of BFHI Authority; and Ghana Breastfeeding Promotion Regulation 2000 (Legislative Instrument [LI] 1667) [18, 19]. Other interventions include Information,

Education and Communication (IEC) materials, and advocacy materials designed for use by health professionals and for the general public. Formulation and implementation of breast-feeding interventions are essential to increase EBF [17]. Health, emotional, cultural, political, and economic issues, as well as other interrelated elements, have an impact on breastfeeding habits such as its initiation and duration [20]. Among these factors, decisions regarding initiation and duration of breastfeeding are influenced by education, employment, place of delivery, family and cultural values [21]. Yet, studies have reported that early initiation of breastfeeding to be associated with EBF [22, 23].

The United Nation's World Food Programme (WFP) implemented the Enhanced Nutrition and Value Chain (ENVAC) intervention [24] which was based on three pillars with pregnant, lactating women, adolescents (10–19 years old) [25] and children under two years old being beneficiaries of the third pillar. The first 1000 days of a child's life from conception to the two years of age are critical for preventing child malnutrition [26]. Lactating women and their children were taken through Social Behavior Change Communication (SBCC) to promote good feeding and care practices [24].

Generational feeding practices are passed on and mothers may be influenced by community and family members' attitudes towards breastfeeding, particularly, EBF [27]. The Enhanced Nutrition and Value Chain, specifically targeting lactating women and their child-pairs with SBCC interventions have the potential to increase exclusive breastfeeding among beneficiaries, Ghana Health Service (GHS) representative were positive about the activities implemented and reported that the intervention had contributed to improved ANC and CWC attendance [24] where mothers are counselled regularly on good infant and young child feeding practices, but has not been assessed yet. In addition, studies on the prevalence of EBF among beneficiaries of the ENVAC programme and its associated factors are limited in Ghana. Therefore, this study assessed the prevalence of EBF practice among lactating mothers with children under two-years old who were beneficiaries of the ENVAC project and its associated factors in northern Ghana.

## Materials and methods

### Study design and area

This was a facility-based study that employed a cross-sectional design. It was conducted in the Sagnarigu Municipality and Tamale Metropolis of the northern region of Ghana. There are 79 communities in the Sagnarigu Municipality. The Municipality is predominantly rural with only 20 and 6 communities classified as urban and peri-urban areas respectively. In the Tamale Metropolis, there are 116 communities with 60 being rural areas and 41 and 15 being urban and peri-urban areas respectively [28]. Agriculture is the main economic activity of the majority of the inhabitants of northern Ghana. They are engaged in both crop and animal farming [11].

Ghana has a three-tiered health system namely primary, secondary and tertiary care. The Community-based Health Planning and Services (CHPS) is the smallest unit of the health system. Other health facilities are health centers, polyclinics, clinics and hospitals run by the government, religious groups and individuals. This constitutes the primary health care level [29]. A health center and district hospital typically serves a population of 20,000 and 100,000–200,000 respectively. All regional hospitals belong to the government at the secondary level of the health system in Ghana. Each regional hospital serves approximately1.2 million people. The teaching and quasi hospitals are part of the tertiary level of the health system in Ghana. These are regarded as complex health care centers of excellence. The Ghanaian government oversees all tertiary health care establishments in the country [30].

## Study population, sample size and sampling

The target population was mothers with children under two years (0–24 months) old who were beneficiaries of the ENVAC project by the World Food Programme (WFP) during their pregnancy and lactating periods. However, only mothers with children aged 6–24 months were included in EBF estimation. These lactating women benefitted from SBCC aimed at improved good feeding and care practices. The SBCC covered the first 1,000 days of life which is a critical period for preventing malnutrition. Health care agents (801) from Ghana Health Service (GHS) in 92 targeted health facilities were given SBCC materials and trained on SBCC; implementation was done at the facility level and through various media especially radio and durbars [24].

The required sample was determined using the estimated proportion of EBF in northern region (33%) [31] and a 5% margin of error and a 95% confidence interval (CI), resulting in 340 mother-child pairs. In addition, six health facilities from the 92 targeted health facilities that were within Sagnarigu Municipal (five facilities) and Tamale Metropolis (one facility) and were primary health care facilities which benefited from the ENVAC project were purposively sampled. Participants from each facility were randomly selected from the various Child Welfare Clinics (CWC) using the CWC registers as the sample frame; interviews were conducted at designated places in the facilities while making sure privacy was ensured. Excel-generated random numbers were used to select participants from CWC registers of the various facilities.

## Data collection and measurement

Data were collected using semi-structured questionnaire through face-to-face interviews for participants who consented to be part of the study. Two research assistants were trained on the questionnaire. In addition, a pre-test of the tool with ten (10) lactating mothers were conducted before main data collection. Data on socio-demographic characteristics such as age of mother, ethnicity, maternal education status, religion and marital status were collected. Other characteristics such as sex of child, age of child and data on child being sick within last two weeks were also obtained. On child feeding characteristics, data on EBF status, initiation of breastfeeding and currently breastfeeding were collected using the infant and young child feeding questionnaire [4]. To assess EBF, all lactating mothers were breastfeeding, so mothers with children who were within six months old were asked if they had given their children any food or drink other than breastmilk or medications and supplements prescribed by certified medical practitioners within the first six months of life [3, 4]. Mothers of children aged six months were also asked the same question following the WHO recommendation for introducing family foods at six months [4] Follow up questions were asked to ascertain whether children were given any prelacteal feeds at birth and, those who responded positively to this question were classified as not practicing EBF.

On maternal education status, the highest education completed was used to group participants into none, moderate and high. Those who had no form of formal education were put into the none category and those who completed primary and junior high school (JHS) were put in the moderate category while those who completed senior high school (SHS) were put in the high category.

## Data analysis

The data were analysed using Statistical Package for Social Sciences (SPSS) software, version 23. The continuous data are presented as means and standard deviations while categorical data is presented as frequencies and percentages. We used Chi-square and Fisher exact test as first line analysis (See S1 File). to identify initial associations and logistic regression was then

performed using forced entry. Variables such as age categories and marital status had larger p-values (>0.2) in the first line analyses and thus were not entered into the multivariable models. EBF prevalence was estimated with a sample size of 327 and used for logistic regression. The goodness of fit test also showed that the logistic regression model was a good fit (Hosmer and Lemeshow test; $X^2 = 6.91$, P = 0.438).

### Ethical consideration

Ethical approval was received from the University for Development Studies (UDS) Institutional Review Board. Permission was also obtained from the various facilities in which the study was conducted. Informed consent was obtained before participants were interviewed. The participants were assured of confidentiality.

## Results

### Socio-demographic and child feeding characteristics of lactating mothers

A total of 339 lactating mothers were interviewed in this study. About a third (32.0%) of the mothers were aged between 18 and 23 years old with the majority (44.0%) aged 24–29 years. One-quarter (25.1%) of them had high education status. The majority (94.4%) of the participants were married. Similarly, the majority (92.6%) of participants had their main source of water being pipe-borne water. More than 4 out of every 5 participants (84.7%) were Muslims. About 18.0% of children were 0–6 months old. The mean age of the children (in months) was 11.9±5.1. The male children were 51.0% (Table 1).

On child feeding practices, 74.6% (95%CI = 69.5–79.2) of children from the programme were exclusively breastfed (Table 2). Nearly all women (98.2%) initiated breastfeeding within the first 30 minutes after delivery. At the time of the study, 72.9% of children were still being breastfed and about 10.0% of children had been sick within the last two weeks before the survey.

### Multivariate analysis of exclusive breastfeeding and sociodemographic factors

The analysis revealed that high maternal education (senior high school or above) and households with access to pipe-borne water were significantly associated with EBF. The amount of variability explained by the variables in this model was 25.4% (Nagelkerke R square = 0.254). Mothers who had moderate or high education status were 4 and 9 times respectively more likely to exclusively breastfeed their children compared to those who had no formal education [aOR = 4.1 (95% CI = 2.18–7.66), P<0.001], and [aOR = 9.15, (95% CI = 3.3–25.4), P<0.001]. Similarly, children who were from households whose main source of water was pipe-borne were about 3 times more likely to exclusively breastfeed as compared to those from households whose main source of water were from wells [aOR = 2.87, (95% CI = 1.1–7.43), P = 0.04] (Table 3).

## Discussion

The current study assessed the prevalence of exclusive breastfeeding (EBF) practice among children under two years in ENVAC project areas and its associated factors in northern Ghana. We found 74.6% of EBF practice in the study setting. High maternal education, and the main source of household drinking water being pipe-borne were the predictors of EBF.

A previous study reported a prevalence of about 70.0% of EBF in southern Ghana [32]. The magnitude is slightly high in the current study. This may be due to the different category of

**Table 1. Socio-demographic characteristics of mothers in study area.**

| Variable | Frequency | Percentage (%) |
|---|---|---|
| **Maternal age categories (Years)** | | |
| 18–23 | 108 | 31.9 |
| 24–29 | 148 | 43.7 |
| 30–35 | 73 | 21.5 |
| ≥ 36 | 10 | 2.9 |
| **Ethnicity** | | |
| Dagomba | 252 | 74.3 |
| Others(Akan, Chokosi, Ewe, Frafra, Fulani, Ga, Gonja, Hausa, Mamprusi and Waala) | 87 | 25.7 |
| **Household water source** | | |
| Well | 25 | 7.4 |
| Pipe-borne water | 314 | 92.6 |
| **Education status** | | |
| None (No formal education) | 132 | 38.9 |
| Moderate (Primary and Junior high school) | 122 | 36.0 |
| High (At least Senior high school) | 85 | 25.1 |
| **Marital status** | | |
| Married | 320 | 94.4 |
| Single | 19 | 5.6 |
| **Religion** | | |
| Christianity | 52 | 15.3 |
| Islam | 287 | 84.7 |
| **Age of child (months)** Mean ± standard deviation | 11.9 ± 5.1 | |
| 0–4 | 12 | 3.5 |
| 5–12 | 185 | 54.6 |
| 13–23 | 142 | 41.9 |
| **Sex of child** | | |
| Female | 166 | 49.0 |
| Male | 173 | 51.0 |

**Table 2. Child breastfeeding characteristics in study area.**

| Variable | Frequency | Percentage (%) |
|---|---|---|
| **Has child been sick within last two weeks** | | |
| No | 304 | 89.7 |
| Yes | 35 | 10.3 |
| **Exclusive breastfeeding status** | | |
| No | 83 | 25.4 |
| Yes | 244 | 74.6 |
| **Initiation of breastfeeding** | | |
| Within 30 minutes | 333 | 98.2 |
| Within an hour | 4 | 1.2 |
| After an hour | 2 | 0.6 |
| **Currently breastfeeding** | | |
| No | 92 | 27.1 |
| Yes | 247 | 72.9 |

**Table 3. Multivariate analysis of factors associated with exclusive breastfeeding (n = 327).**

| Variables | Adjusted Odds Ratio (aOR) | 95% C.I. | | P-value |
|---|---|---|---|---|
| | | Lower | Upper | |
| **Child sex** | | | | |
| Male | Reference | | | |
| Female | 1.41 | 0.81 | 2.52 | 0.208 |
| **Ethnicity** | | | | |
| Dagomba | Reference | | | |
| Others | 0.72 | 0.3 | 1.74 | 0.465 |
| **Educational status** | | | | |
| None | Reference | | | |
| Moderate | 4.1 | 2.18 | 7.66 | <0.001 |
| High | 9.15 | 3.3 | 25.4 | <0.001 |
| **Religious affiliation** | | | | |
| Islam | Reference | | | |
| Christianity | 1.71 | 0.4 | 7.2 | 0.467 |
| **Household's main source of water** | | | | |
| Well | Reference | | | |
| Pipe-borne water | 2.87 | 1.1 | 7.43 | 0.029 |

participants, the current participants are from the ENVAC project areas in the northern region where women were taken through social behavior change communication to improve feeding and care practices. Recent national estimates of EBF have been lower (52.0% in 2014 vs 42.9% in 2018) than the prevalence reported in this study [11, 12]. In our study, the high prevalence of EBF could be due to increased maternal knowledge through infant and young child feeding practices (IYCF) education at CWCs [32]. Thus, these participants are different from community level members. However, some developed countries have reported lower prevalence of EBF (13.8% - 62.0%) than the current findings [33, 34]. A study conducted in Australia found about nine out of ten children being exclusively breastfed at birth, this decreased to 60.0% within the early postnatal visit (24-hours prior to first postnatal health visit) [34]. This sharp decline is likely due to cultural practices [35], availability of infant formulas [36] and exposure to messages on IYCF [37].

In our study, mothers with high education (senior high school or above) were more likely to exclusively breastfeed their children compared to those who had no formal education. This finding is similar to other studies where higher education was associated with increased likelihood of breastfeeding exclusively. Though, mothers with high education are likely to have occupations that may be time-demanding and affect EBF practices, we did not find such patterns in the current study. Besides, higher education may enhance caregivers' understanding and uptake of nutrition education on infant and young child feeding practices. A study conducted in Ethiopia reported that the main reason for discontinuation of EBF is mothers returning to work, the authors suggested that a revised national policy on maternity leave to six-months could improve the prevalence of EBF [38]. Similarly, a study conducted in Ghana found that working mothers have the tendency of not practicing breastfeeding up to six months due to the short maternity leave of three months [15] and yet had the least prevalence of stunting in their children [39]. A trial conducted in Norway showed that Baby Friendly Initiative in community health service increased the magnitude of EBF up to six months [40]. This is an indication that other factors may affect a mother's odds of exclusive breastfeeding. Another study conducted in Belgium demonstrated a positive association of maternal education on breastfeeding [41].

Our study revealed that children from households with pipe-borne water as their main source of water were more likely to be exclusively breastfed compared to those whose main source of water were from wells. In low-income settings, it is often the responsibility of women to make water available in the home. This is one of the gender roles of women in the study setting [42, 43]. Thus, women including lactating mothers likely spent a significant amount of their time fetching water from wells or boreholes in addition to other household responsibilities which increase their daily workload. In the study setting, extended family support for child care is high. While lactating mothers may be engaged in other family chores including water provision, other family members often help with child care. While this has advantages to the mother by reducing the additional burden on mothers, extended family caregivers (especially grandmothers) are more likely to give or introduce food and water to console hungry and crying children in the absence of their mothers. According to Lau (2001), lactation insufficiency is attributable to maternal stresses [44]. Hence, the synergistic effect of increased workload cum stress and the anxiety of having to ensure water security while performing other household responsibilities could possibly lead to lactation insufficiency. Another study demonstrated that stress and anxiety are important biological factors potentially associated with lactogenesis [45]. Additionally, water from unprotected wells may be unsafe due to contamination. Children may be infected with diarrhea conditions resulting in the introduction of other foods and liquids before six months of age [46].

The strength of this study is that it provides data on EBF among beneficiaries of the ENVAC project and its associated factors. It contributes to available information on effects of supplementation programmes and nutrition enhancement interventions for advocacy among marginalised populations. However, the study has several limitations. The study did not measure cultural practices which may be critical in the determination of EBF. Moreover, the amount of variability explained by the variables in the EBF model was 25.4% indicating that other important variables were not measured in the current study. Social desirability on the part of mothers could have caused an overestimation of the magnitude of exclusive breastfeeding, especially since participants had received SBCC messages and knew what they were supposed to do to meet EBF target and could easily recall it to interviewers despite significant interviewer training to ensure that questions were asked in a non-judgmental way. Also interviews were conducted in health facilities which could represent a significant selection bias where the most educated or enthusiastic mothers are more likely to attend.

In addition, the required sample size for this study was 340 but 327 was used to estimate the EBF prevalence which could have affected the power of the study. Nonetheless, a post hoc power analysis showed that the impact was likely minimal. Finally, though participants of this study were beneficiaries of the ENVAC project, there was no control group/ area to be able to determine if high EBF prevalence was as a result of the project or other factors, hence one cannot be certain that the high EBF is attributable to the ENVAC project. Nevertheless, Ghana's multiple indicator cluster survey showed that more women from rural areas (45.8%) practice EBF as compared to women from urban areas (38.7%) [8], yet, this current study reports an EBF prevalence of 74.6% in participants who were largely been from urban and peri-urban areas.

## Conclusion

A social behaviour change communication strategy implemented by ENVAC to lactating mothers likely improved exclusive breastfeeding practice in two districts of northern Ghana. EBF practices were higher among beneficiaries with high education and households with access to pipe-borne water. Our results demonstrate the potential for a combined SBCC

strategy along with key maternal and household level factors to increase exclusive breastfeeding rates in impoverished communities such as those we studied as part of the ENVAC study in northern Ghana. This means that future studies that seek to improve exclusive breastfeeding among lactating mothers through SBCC strategies need to also address such factors as maternal education and access to safe and clean water for maximum impact.

## Supporting information

**S1 File.**
(DOCX)

## Author Contributions

**Conceptualization:** Martin Nyaaba Adokiya, Mohammed Bukari, Amata Atinlie Amoasah.

**Data curation:** Martin Nyaaba Adokiya, Mohammed Bukari, Humphrey Garti, Matthew Y. Konlan, Zakari Ali.

**Formal analysis:** Martin Nyaaba Adokiya, Mohammed Bukari.

**Investigation:** Martin Nyaaba Adokiya, Joyce A. Ndago, Robert B. Kuganab-Lem, Zakari Ali.

**Methodology:** Martin Nyaaba Adokiya, Joyce A. Ndago, Robert B. Kuganab-Lem.

**Resources:** Martin Nyaaba Adokiya.

**Supervision:** Martin Nyaaba Adokiya, Robert B. Kuganab-Lem, Humphrey Garti, Matthew Y. Konlan.

**Validation:** Martin Nyaaba Adokiya, Matthew Y. Konlan, Amata Atinlie Amoasah, Zakari Ali.

**Writing – original draft:** Martin Nyaaba Adokiya, Mohammed Bukari, Zakari Ali.

**Writing – review & editing:** Martin Nyaaba Adokiya, Joyce A. Ndago, Robert B. Kuganab-Lem, Humphrey Garti, Matthew Y. Konlan, Amata Atinlie Amoasah, Zakari Ali.

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
