## [Decision Letter · Decision Letter 0]

23 Feb 2023

PONE-D-23-01016Exclusive breastfeeding among beneficiaries of a nutrition enhancement programme and its associated factors in GhanaPLOS ONE

Dear Dr. Adokiya,

Thank you for submitting your manuscript to PLOS ONE. After careful consideration, we feel that it has merit but does not fully meet PLOS ONE’s publication criteria as it currently stands. Therefore, we invite you to submit a revised version of the manuscript that addresses the points raised during the review process.

We look forward to receiving your revised manuscript.

Kind regards,

Neetu Choudhary, PhD

Academic Editor

PLOS ONE

Journal Requirements:

Reviewers' comments:

Reviewer's Responses to Questions

**Comments to the Author**

1. Is the manuscript technically sound, and do the data support the conclusions?

Reviewer #1: Partly

Reviewer #2: Yes

2. Has the statistical analysis been performed appropriately and rigorously? 

Reviewer #1: I Don't Know

Reviewer #2: Yes

3. Have the authors made all data underlying the findings in their manuscript fully available?

Reviewer #1: Yes

Reviewer #2: Yes

4. Is the manuscript presented in an intelligible fashion and written in standard English?

Reviewer #1: Yes

Reviewer #2: Yes

5. Review Comments to the Author

Reviewer #1: The study is important and will be of interest to an international audience. The written text would benefit from academic mentoring to strengthen unsupported claims and add substance to the conclusion ( rather than repeating the results).

Statements need supporting from the evidence unless they are common knowledge for example, in the introduction, the points on boost to immune system, 36% global EBF, and Australia's EBF at 96% are all without citations to support the claim and the entire manuscript needs to be revised for this point. See also p13 A study in Australia found 9/10 children EBF at birth, no citation.

Citation 12 is used to emphasize initiation of breastfeeding on the rise in Nigeria, yet the date of publication is 2010 and initiation does stray from the main point of EBF and this should be brought out in the text and not left to the reader to work it through.

'Adolescent' needs to be defined by age as used in this study ( and supported by literature) for an international audience.

It needs to be stated if the ENVAC project captured data on prevalence of EBF, as the readers may consider the point.

Study design, a diagram of the regions and numbers and levels of subgroups in the areas would be helpful for readers to visualize the communities.

The important point is not to repeat what has already been stated well once. There is repetition on the mother child pairs which could be trimmed back.

The use of the word 'targeted' on p7 sounds as though the participants were not voluntary participants, perhaps the word 'invited' could be used instead.

Please describe the purposive sampling exactly and why this was necessary, and the random sampling and how this was done exactly.

How was data collected, was it only by interview, that is all that they consented to?

The definition of EBF in some studies includes no medicine, have you made your position clear on this ? P8

When reporting statistics it is preferable to use the same number of decimal places throughout see p9 68% v 92.6% (2 v 3 places).

P10 mean age of children needs to be expressed as 'months'.

Child not being sick as a predictor of EBF is rather a confusing way to report what is occurring the other way around- EBF a predictor that a child wont be sick, if other triggers are controlled for. The authors need to add to the conclusion on this.

Reviewer #2: The current study looked at the exclusive breastfeeding rate (EBF) and its determinants among beneficiaries of a WFP-sponsored programme in Ghana's northern region. The authors reported a high prevalence of EBF and identified maternal education, a sickness episode in the previous two weeks, and the source of household drinking water as important determinants of EBF in the study setting. Overall, this is an important topic for the study setting. The study design and analytical techniques are sound in order to achieve the study's objectives. The manuscript is also well written in standard English. However, there are a few issues that must be addressed before the manuscript can be considered for publication in PLOS ONE.

General observations: The manuscript requires extensive copyediting to correct grammar issues such as incorrect sentence structure, missing punctuation or articles, and inconsistent font size. A conceptual framework should also guide the variables included as potential predictors, as some do not have a scientific basis. The following are some specific remarks:

Abstract

When you use including, you want to show that there are many but only highlight a few. When the exact number of pillars is specified, the word namely is more appropriate.

Make the study's purpose more specific; what about EBF was measured? Furthermore, the goal is stated as if the EBF rate was determined among women and children. The entire purpose sentence should be revised.

The sentence “Participants were lactating women who benefitted from the ENVAC

project using Social Behavior Change Communication (SBCC) and facilitated access to

Complementary Nutritious Foods (CNF) through market and vouchers.” is confusing

… those whose children or child?

Introduction

The introduction is well-written, and the problem is clearly stated and justified. Nonetheless, the authors should ensure that the manuscript is thoroughly copyedited; there are missing punctuations and font sizes that do not match other text within the manuscript.

Methods

Authors may have to redefine the study’s target population for clarity

The study does not state whether the six purposefully chosen facilities were distributed evenly between the two areas, Sagnarigu and Tamale Metropolitan. It also does not specify how the overall sample was distributed among the various facilities chosen.

Outcome variable: more information is needed to define the outcome variable properly. What was the recall period for giving breastmilk or foods?

Exposure variable: I am surprised by the study variables' conceptualization, including the use of sickness in the previous two weeks as a potential predictor variable. EBF primarily affects children in their first six months of life. As a result, it will be interesting to learn from the current study that an exposure after 6 months predicts an outcome that occurred long before the exposure.

Results

The sample size needed to achieve the study's objectives was 340. 339 people were interviewed, however. Given that no adjustments were made to the sample size to account for non-response, not interviewing all 340 women has implications for generalising the findings. That should be addressed adequately.

Table 1: The mean age of the children can be placed alongside their age variable.

The categorisation of the timing of initiation of breastfeeding is problematic. Any child who was breastfed within 30 minutes is still eligible for the within 1 hour group. That should be reconsidered. Furthermore, did all of the women say they had ever breastfed? When the denominator is clearly indicated or defined, the EBF rate is easier to understand and interpret.

Multivariate analysis: “… household main source of water of being pipe-borne were significantly associated with EBF” what does it mean?

Furthermore, the narrative discusses the factors associated with EBF in general, so there is no need to attempt to indicate the direction when it is captured in the sentence following.

What method was used to fit the multivariable model? Table 3 lacks variables such as maternal age categories and marital status. What factors influenced the exclusion of these variables?

The section on data analysis mentioned comparison tests. I am curious what role they played in the current study.

Discussion

“A previous study reported a prevalence of about 70.0% of EBF in southern Ghana [24]. Though, there are geographical differences between northern and southern Ghana.” means?

The explanation for the difference in prevalence rate between health facility-based designs and population-based designs is not entirely justifiable. The authors contend that the participants recruited at the health facility are distinct from community members. The question is where did those women come from?

Similarly, the explanation for the link between water source and EBF may be implausible. Given the authors' explanation, I am wondering if the EBF rates will differ between women who fetch water from pipped sources and those who fetch from wells, assuming the travel distance is the same or similar.

The current study's design and limited number of exposure variables should be viewed as significant limitations by the authors.

Conclusion: check error

6. PLOS authors have the option to publish the peer review history of their article (what does this mean?). If published, this will include your full peer review and any attached files.

Reviewer #1: No

Reviewer #2: **Yes: **Dr Michael Boah

---

## [Author Response · Author response to Decision Letter 0]

20 Apr 2023

Response to reviewers

We are grateful for the helpful comments and we have adequately addressed them. 

Comments Responses

Reviewer 1 

Comment: The study is important and will be of interest to an international audience. The written text would benefit from academic mentoring to strengthen unsupported claims and add substance to the conclusion ( rather than repeating the results). Response: We thank the reviewer for the comment. We have made a thorough revision of the paper and added references where required. We have also re-written the conclusion to derive from the study. 

Comment: Statements need supporting from the evidence unless they are common knowledge for example, in the introduction, the points on boost to immune system, 36% global EBF, and Australia's EBF at 96% are all without citations to support the claim and the entire manuscript needs to be revised for this point. See also p13 A study in Australia found 9/10 children EBF at birth, no citation. Response: We have revised the manuscript to include missing citations. 

Comment: Citation 12 is used to emphasize initiation of breastfeeding on the rise in Nigeria, yet the date of publication is 2010 and initiation does stray from the main point of EBF and this should be brought out in the text and not left to the reader to work it through. Response: Thank you for drawing our attention to the discrepancy. We have revised this section appropriately.

Comment: 'Adolescent' needs to be defined by age as used in this study (and supported by literature) for an international audience. Response: We have revised and included the age of adolescents. We used WHO definition and have also provided the appropriate reference (Page 6, lines 40). 

Comment: It needs to be stated if the ENVAC project captured data on prevalence of EBF, as the readers may consider the point. Response: We have revised the introduction to addresses this concern (Page 7, lines 46-51)

Comment: Study design, a diagram of the regions and numbers and levels of subgroups in the areas would be helpful for readers to visualize the communities. Response: We did not address this comment. We think that this information may not make difference. 

Comment: The important point is not to repeat what has already been stated well once. There is repetition on the mother child pairs which could be trimmed back. Response: We thank the reviewer for noting this, we have made extensive revisions throughout the paper to trim redundancies.

Comment: The use of the word 'targeted' on p7 sounds as though the participants were not voluntary participants, perhaps the word 'invited' could be used instead. Response: We have revised the sentence to remove the word “targeted” and used “invited” as suggested by the reviewer to avoid ambiguity.

Comment: Please describe the purposive sampling exactly and why this was necessary, and the random sampling and how this was done exactly. Response: We have made revisions to improve on the description of the sampling methods. See methods (page 8-9, lines 87-89). 

Comment: How was data collected, was it only by interview, that is all that they consented to? Response: Yes, data were collected by interview only. We have stated this clearly in the methods section (page 9, lines 95-96). 

Comment: The definition of EBF in some studies includes no medicine, have you made your position clear on this? P8 Response: We followed the standard WHO definition of EBF which includes medicines prescribed by a medical practitioner

Comment: When reporting statistics it is preferable to use the same number of decimal places throughout see p9 68% v 92.6% (2 v 3 places). Response: We thank the reviewer for noting this and have revised to have consistent number of decimal places

Comment: P10 mean age of children needs to be expressed as 'months'. Response: This has been revised appropriately

Comment: Child not being sick as a predictor of EBF is rather a confusing way to report what is occurring the other way around- EBF a predictor that a child won’t be sick, if other triggers are controlled for. The authors need to add to the conclusion on this.

 Response: We have revised this after a careful reconsideration.

Reviewer 2 

Comment: The current study looked at the exclusive breastfeeding rate (EBF) and its determinants among beneficiaries of a WFP-sponsored programme in Ghana's northern region. The authors reported a high prevalence of EBF and identified maternal education, a sickness episode in the previous two weeks, and the source of household drinking water as important determinants of EBF in the study setting. Overall, this is an important topic for the study setting. The study design and analytical techniques are sound in order to achieve the study's objectives. The manuscript is also well written in standard English. However, there are a few issues that must be addressed before the manuscript can be considered for publication in PLOS ONE.

 Response: We thank the reviewer for agreeing to review our work and we have addressed all the comments raised herein.

Comment: General observations: The manuscript requires extensive copyediting to correct grammar issues such as incorrect sentence structure, missing punctuation or articles, and inconsistent font size. A conceptual framework should also guide the variables included as potential predictors, as some do not have a scientific basis. The following are some specific remarks: Response: We thank the reviewer for their comments. We have extensively revised the manuscript to correct language errors. In addition, variables included in the model were examined in a first line analysis using Chi-square and Fisher exact test before variables were taken further for logistic regression file. An additional file containing the results for the preliminary bivariate analyses are included in the online additional files.

Abstract

Comment: When you use including, you want to show that there are many but only highlight a few. When the exact number of pillars is specified, the word namely is more appropriate. Response: We have noted this and have applied the revision where necessary.

Comment: Make the study's purpose more specific; what about EBF was measured? Furthermore, the goal is stated as if the EBF rate was determined among women and children. The entire purpose sentence should be revised.

 Response: The purpose statement has been revised appropriately.

Comment: The sentence “Participants were lactating women who benefitted from the ENVAC project using Social Behavior Change Communication (SBCC) and facilitated access to Complementary Nutritious Foods (CNF) through market and vouchers.” is confusing … those whose children or child? Response: The sentence has been revised.

Introduction

Comment: The introduction is well-written, and the problem is clearly stated and justified. Nonetheless, the authors should ensure that the manuscript is thoroughly copy-edited; there are missing punctuations and font sizes that do not match other text within the manuscript.

 Response: We have taken note of this and made the necessary revisions. 

Methods

Comment: Authors may have to redefine the study’s target population for clarity. The study does not state whether the six purposefully chosen facilities were distributed evenly between the two areas, Sagnarigu and Tamale Metropolitan. It also does not specify how the overall sample was distributed among the various facilities chosen. Response: More information has been added on the study’s target population to improve clarity and information has also been included on how facilities were distributed as well as how the overall sample was distributed (See additional file, Table 1)

Comment: Outcome variable: more information is needed to define the outcome variable properly. What was the recall period for giving breastmilk or foods? Response: We have revised to include the recall period (Page 9, lines 103-107).

Comment: Exposure variable: I am surprised by the study variables' conceptualization, including the use of sickness in the previous two weeks as a potential predictor variable. EBF primarily affects children in their first six months of life. As a result, it will be interesting to learn from the current study that an exposure after 6 months predicts an outcome that occurred long before the exposure. Response: We have carefully reconsidered this observation and agree with the reviewer, hence, we have removed child sickness from the models. 

Results

Comment: The sample size needed to achieve the study's objectives was 340. 339 people were interviewed, however. Given that no adjustments were made to the sample size to account for non-response, not interviewing all 340 women has implications for generalising the findings. That should be addressed adequately. Response: We have acknowledged this a limitation of the study (Page 18, lines 226-228) 

Comment: Table 1: The mean age of the children can be placed alongside their age variable. Response: We have placed mean age alongside the age variable, thank you.

Comment: The categorisation of the timing of initiation of breastfeeding is problematic. Any child who was breastfed within 30 minutes is still eligible for the within 1 hour group. That should be reconsidered. Response: We agree that any child who was breastfed within 30 minutes is still eligible for the within 1 hour group. The responses were grouped into 30 minutes, within 1 hour and after 1 hour but were collapsed during the analysis, we have revised to retain the initial grouping

Comment: Furthermore, did all of the women say they had ever breastfed? When the denominator is clearly indicated or defined, the EBF rate is easier to understand and interpret.

 Response: Yes, all the women said they were breastfeeding.

Comment: Multivariate analysis: “… household main source of water of being pipe-borne were significantly associated with EBF” what does it mean? Response: We intended to say that households with access to pipe-borne water were more likely to practice EBF, the sentence has been revised for more clarity (page 13, lines 149-150).

Comment: Furthermore, the narrative discusses the factors associated with EBF in general, so there is no need to attempt to indicate the direction when it is captured in the sentence following. Response: We thank the reviewer for noting this, various revisions have been made where necessary. 

Comment: What method was used to fit the multivariable model? Table 3 lacks variables such as maternal age categories and marital status. What factors influenced the exclusion of these variables? Response: We used Chi-square and Fisher exact test as first line analysis to identify initial associations and logistic regression was then performed using forced entry. Variables such as age categories and marital status had larger p-values (>0.2) in the first line analyses and thus were not entered into the multivariable models. 

Comment: The section on data analysis mentioned comparison tests. I am curious what role they played in the current study. Response: We have revised to clarify. 

Discussion

Comment: “A previous study reported a prevalence of about 70.0% of EBF in southern Ghana [24]. Though, there are geographical differences between northern and southern Ghana.” means? Response: The statement “Though, there are geographical differences between northern and southern Ghana.” Has been deleted to improve clarity.

Comment: The explanation for the difference in prevalence rate between health facility-based designs and population-based designs is not entirely justifiable. The authors contend that the participants recruited at the health facility are distinct from community members. The question is where did those women come from?

 Response: We agree with the reviewer’s observation and have revised appropriately (Page 15, lines 166-170)

Comment: Similarly, the explanation for the link between water source and EBF may be implausible. Given the authors' explanation, I am wondering if the EBF rates will differ between women who fetch water from pipped sources and those who fetch from wells, assuming the travel distance is the same or similar.

 Response: We have revised to improve clarity (Page 16-17, lines 198-205)

Comment: The current study's design and limited number of exposure variables should be viewed as significant limitations by the authors. Response: We have included this as a limitation in the discussion section.

Comment: Conclusion: check error

 Response: We have revised the conclusion appropriately.

---

## [Editor Report · Decision Letter 1]

4 May 2023

PONE-D-23-01016R1Exclusive breastfeeding among beneficiaries of a nutrition enhancement programme and its associated factors in GhanaPLOS ONE

Dear Dr. Adokiya,

Thank you for submitting your manuscript to PLOS ONE. After careful consideration, we feel that it has merit but does not fully meet PLOS ONE’s publication criteria as it currently stands. Therefore, we invite you to submit a revised version of the manuscript that addresses the points raised during the review process.

cols: These article types are not expected to include results but may include pilot data. 

We look forward to receiving your revised manuscript.

Kind regards,

Neetu Choudhary, PhD

Academic Editor

PLOS ONE

Journal Requirements:

Additional Editor Comments:

Thank you for your revisions! However, the conclusion still needs revision. Please note the conclusion is not just summary of results. You can include key observations with some future implications.

---

## [Author Response · Author response to Decision Letter 1]

16 May 2023

Comment: Thank you for your revisions! However, the conclusion still needs revision. Please note the conclusion is not just summary of results. You can include key observations with some future implications. 

Response: We thank the Editor for the additional comments. We have revised the conclusion session of the abstract and main manuscript including possibly future implications.

---

## [Editor Report · Decision Letter 2]

18 May 2023

Exclusive breastfeeding among beneficiaries of a nutrition enhancement programme and its associated factors in Ghana

PONE-D-23-01016R2

Dear Dr. Adokia,

We’re pleased to inform you that your manuscript has been judged scientifically suitable for publication and will be formally accepted for publication once it meets all outstanding technical requirements.

Kind regards,

Neetu Choudhary, PhD

Academic Editor

PLOS ONE
---

## [Editor Report · Acceptance letter]

19 May 2023

PONE-D-23-01016R2 

Exclusive breastfeeding among beneficiaries of a nutrition enhancement programme and its associated factors in Ghana 

Dear Dr. Adokiya:

I'm pleased to inform you that your manuscript has been deemed suitable for publication in PLOS ONE. Congratulations! Your manuscript is now with our production department. 

Kind regards, 

on behalf of

Dr. Neetu Choudhary 

Academic Editor

PLOS ONE